# Local Dependency-Enhanced Graph Convolutional Network for Aspect-Based Sentiment Analysis

**Fei Wu** [†] [iD] **and Xinfu Li** [*,†]

Cyberspace Security and Computer College, Hebei University, Baoding 071000, China; wufei@stumail.hbu.edu.cn
* Correspondence: mc_lxf@126.com
† These authors contributed equally to this work.

**Abstract:** The task of aspect-based sentiment analysis (ABSA) is to detect the sentiment polarity toward given aspects. Contemporary methods predominantly utilize graph neural networks and incorporate attention mechanisms to dynamically connect aspect terms with their surrounding contexts, resulting in more informative feature representations. However, these methods only consider whether there are dependencies between words when introducing dependencies, ignoring that dependencies between different sentiment words have different effects. Neglecting this could introduce noise and negatively impact the model's performance. To overcome this limitation, we introduce a novel approach called the local dependency-enhanced graph convolutional network (LDEGCN). Our method combines semantic information and dependency relationships to better capture the affective relationships between words. Specifically, we integrate sentiment knowledge from SenticNet to enrich the sentence's dependency graph and thoroughly explore the dependency types between contexts and aspects to focus on particular dependency types. The local context weight (LCW) method is employed on the dependency-enhanced graph to emphasize the importance of local contexts, thereby mitigating the issue of long-distance dependencies. Through extensive evaluations of five public datasets, the LDEGCN model demonstrates significant improvements over mainstream models.

**Keywords:** aspect-based sentiment analysis; graph convolutional network; multi-head attention; affective knowledge; dependency types

## 1. Introduction

With the continuous advancement of Internet technology and the growing number of Internet users, a substantial volume of comment data is being generated through information exchange. These valuable comment data, such as reviews on events, products and people, can be leveraged for extracting useful information through sentiment analysis [1]. However, traditional sentence-level sentiment analysis methods have gradually become inadequate in meeting our requirements, as users often pay attention to multiple aspects of the same entity [2]. As illustrated in Figure 1, the sentiment polarity of the aspect "environment" is considered positive, whereas the sentiment polarity of the aspect "food" is negative. Consequently, ABSA has emerged as a crucial research area in the field of natural language processing (NLP) [3,4].

The key to addressing the ABSA task lies in precisely identifying the opinion words that hold a substantial influence in determining the sentiment polarity of each aspect [5]. In early stages, ABSA tasks employed a combination of recurrent neural networks (RNNs) and attention mechanisms [6–10]. These approaches were used to capture aspect-related semantic information and generate aspect-specific sentence representations. However, these methods suffer from limitations when it comes to handling noise introduced by unrelated words. Additionally, they overlook the crucial syntactic dependency information

within sentences, and the attention module might unintentionally emphasize irrelevant words due to syntactic omissions.

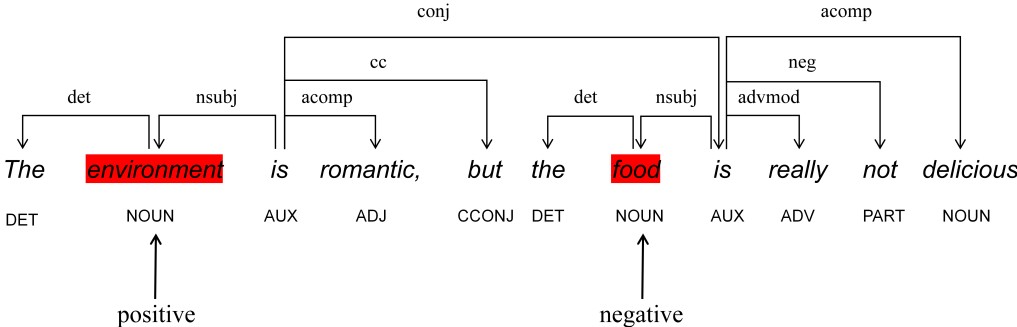

**Figure 1.** This is an example sentence with a dependency tree, in which aspect terms (highlighted in red) are connected with other words according to their syntactic dependency.

Recently, models have been developed with the objective of establishing relational connections between aspects and their corresponding opinion words [11–16]. These methods notably improve the model's performance, while they treat all dependencies equally, disregarding the fact that the impacts of dependencies among distinct sentiment words differ significantly in ABSA tasks. From a linguistic perspective, dependencies between words with different sentiment intensities carry distinct meanings. Words exhibiting strong sentiment orientation can play a significant role in ABSA tasks, while words with ambiguous sentiment tendencies may interfere with the model's judgment. From a syntactic perspective, different dependency types have varying levels of importance in sentiment analysis tasks. Tian et al. [17] argue that dependencies of the "nsubj" (nominal subject) type, as well as certain modifier dependencies like "amod" (adjective modifier) and "advmod" (adverb modifier), are more crucial than other types of dependencies.

To address the aforementioned limitations, we introduce the LDEGCN model, which is composed of three crucial modules: the semantic feature extraction module, the dependency-enhanced module and the local context weight (LCW) method. The semantic feature extraction module is designed to extract meaningful and contextually relevant features from the output of the embedding layer. It consists of two layers of multi-head attention to assign attention weights to aspects within the context, thereby facilitating the extraction of semantic features. The dependency-enhanced module is utilized to strengthen the dependency relationships between words within a sentence. It focuses on specific dependency relationships by integrating external affective knowledge and dependency types. Specifically, we first construct a dependency graph based on the dependency tree. To harness external sentiment knowledge and explore various dependency types, we utilize SenticNet to assign intensity scores to sentiment words and construct a specific dependency set, $S_{set}$. As a result, the enhanced graph can effectively capture the emotional connections and dependency relationships between contexts and aspects. Furthermore, the LCW method [18] is employed on the dependency-enhanced graph to weight the dependencies, effectively solving the problem of long-distance dependencies. By evaluating our method on five benchmark datasets, we have observed significant improvements compared to mainstream models.

The main contributions of this paper can be summarized as follows:

- We propose an aspect-aware mechanism based on multi-head interactive attention and multi-head self-attention to enhance the representation of aspect-related semantic features.
- We utilize SenticNet for graph construction to introduce external sentiment knowledge and enhance the focus on specific dependency relationships within the graph by building a specific set of dependencies.
- The LCW method is employed on the dependency-enhanced graph, which effectively diminishes attention on long-distance dependencies.

## 2. Related Works

In recent years, sentiment analysis has witnessed remarkable success with the adoption of deep learning models. These models have the capability to automatically learn valuable features from raw text data and encode sentences using low-dimensional word vectors, thereby capturing rich semantic information. The previous related work in this field can be categorized into three different parts:

### 2.1. Attention-Based models

Several attention-based networks have demonstrated promising performance by implicitly modeling the semantic relation of an aspect and its context. Chen [7] introduced a multiple-attention-based memory network to capture crucial information pertaining to sentiment orientation for different aspect words. Ma et al. [8] introduced a pair of attention networks to dynamically learn representations for both aspects and contexts. Huang et al. [9] presented the attention-over-attention (AOA) module to simultaneously learn representations for both aspects and contexts. Fan et al. [10] exploited a multiple fine-grained attention model to interactively learn the relations between aspects and their contexts.

Although these models benefit from attention and syntactic information, they fail to capture dependency relationships between contexts, which is of paramount importance for ABSA tasks.

### 2.2. Graph Neural Networks

Due to the enhanced capability of graphs in representing the structural information of text and accurately capturing word-level dependency information, researchers have begun to explore the utilization of graph neural networks (GNNs) for learning feature representations from dependency trees. Zhang and Sun [11,12] presented a graph convolutional network (GCN) method to learn from the node representation of the dependency tree. Liang et al. [13] proposed a method to construct a syntactical dependency graph for a sentence based on a specific aspect. It introduces the interactive graph convolutional networks (InterGCNs) model to extract aspect-focused and inter-aspect sentiment features for a particular aspect. Wang et al. [14] introduced a relation graph attention network (R-GAT) that constructs a node relation graph within the dependency tree. It employs graph attention mechanisms to capture complex dependencies between nodes and aggregates node information into a graph-level representation. Li et al. [15] proposed a DualGCN model to simultaneously consider the complementary interaction between syntactic structure and semantic relevance.

With the increasing number of GNN-based models demonstrating superior performance, research has proven the effectiveness of enhancing the dependencies between contexts and aspects. However, these models treat each dependency item equally and they ignore the affective information within contexts and aspects.

### 2.3. Affective Knowledge

By integrating external data and knowledge resources, NLP models can gain a deeper understanding and processing capability for natural language, leading to substantial advancements across various domains and fields [19,20]. In sentiment classification tasks, sentiment knowledge significantly improves the model's understanding and expression of the emotional content in text [21]. Zhou et al. [22] introduced a commonsense knowledge graph to enhance sentence representation. Xing et al. [23] adapted various existing sentiment lexicons to the target domain and found that SenticNet, as a universal sentiment lexicon, outperformed other lexicons in terms of performance. In ABSA tasks, leveraging the sentiment lexicon SenticNet allows for the extraction of aspect-dependent sentiment expressions from the context. Liang et al. [24] injected affective knowledge into the graph to enhance the model's ability to identify the emotional tendencies in the text, leading to more accurate classification predictions.

Overall, infusing affective knowledge is a promising research direction that enables the fusion of external resources with neural network models. This integration empowers the model to better comprehend the emotional information in the text and enriches the feature representation.

## 3. Proposed Approach/ Methodology

In this part, we introduce a comprehensive description of LDEGCN. This approach aims to address the challenges of capturing syntactic and semantic information in ABSA tasks. It enhances the representation of aspect-related semantic features through a multi-layer attention structure. The advantages of GNN and dependency tree structure are fully exploited, combining external sentiment knowledge and dependency type information to bolster the representation of relevant dependencies in the dependency graph. The overall framework of the network is illustrated in Figure 2. The proposed method is mainly divided into five components: embedding layer, semantic feature extraction module, LDEGCN module, feature fusion layer and sentiment classifier.

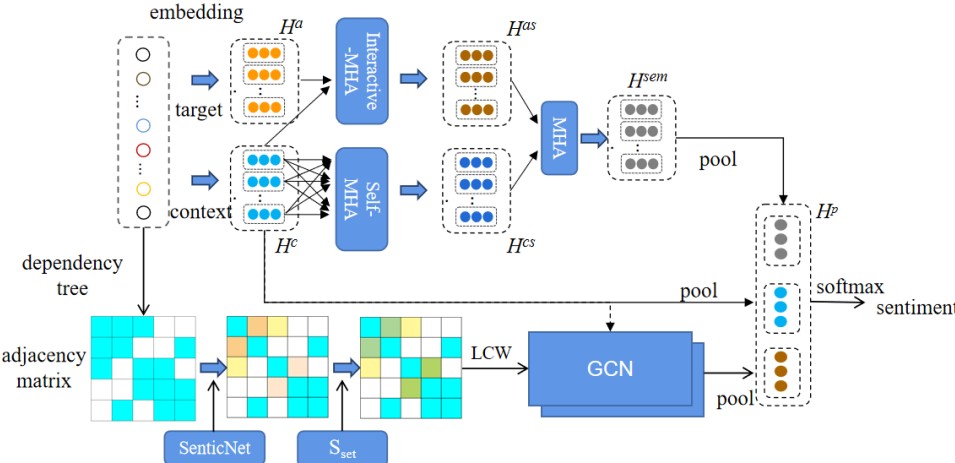

**Figure 2.** Framework of the proposed LDEGCN model.

### 3.1. Task Definition

Given a context sequence, $S = \{w_1, w_2, \ldots, w_{a+1}, \ldots, w_{a+m}, \ldots, w_n\}$, of length $n$, where $A = \{a_1, a_2, \ldots, a_m\}$ is the aspect sequence and the subsequence of the sentence, $S$. A sentence can contain one or multiple aspect terms corresponding to three sentiment polarities (positive, negative and neutral). ABSA aims to detect the sentiment polarity of the given aspect, $A$, in a sentence, $S$, by extracting sentiment information from the context.

### 3.2. Embedding

In ABSA tasks, pre-trained language model BERT offers rich feature information. Therefore, we input the sentence-aspect pair $(S, A)$ into BERT in the form of "$[CLS]$ $S$ $[SEP]$ $A$ $[SEP]$" to initialize the aspect-aware word vectors, where "$[CLS]$" represents the symbolic marker encoding the overall sentence-level representation, while "$[SEP]$" serves as the separator between the context and the aspect. The calculation in BERT is as follows:

$$\{H^c, H^a\} = \text{BERT}(\{[CLS], S, [SEP], A, [SEP]\}) \tag{1}$$

In the equation, $H^c = \{h_1, h_2, \ldots, h_n\} \in R^{n \times h_{dim}}$ is the word embedding of contextual word and $H^a = \{h_{a+1}, h_{a+2}, \ldots, h_{a+m}\} \in R^{m \times h_{dim}}$ is the word embedding of aspect word, where $h_{dim}$ is the dimensionality of each word vector, $n$ is the length the sentence and $m$ is the length of the aspect term.

### 3.3. Semantic Feature Extraction

Attention is a commonly used method to grasp the interaction between aspects and the surrounding contexts [25]. The semantic feature extraction module comprises a two-layer multi-head attention mechanism. Multi-head attention (MHA) is a variation of attention. The input sequence undergoes multiple linear transformations and is mapped to several vector representations, including query, key and value. Each new query and key–value pair undergoes an independent attention operation, referred to as a head. The attention values generated by different heads are concatenated to obtain the final output value. The mathematical calculation is as follows:

$$\text{Attention}(K, Q) = \text{softmax}\left( \frac{QW^Q \times \left(KW^K\right)^{\text{T}}}{\sqrt{d}} \right) K \tag{2}$$

where matrices $Q$ and $K$ are the query vector and the key vector, respectively, while $W^Q$ and $W^K$ are learnable weight matrices and $d$ is the dimensionality of the input node features.

$$\text{MHA}(K, Q) = \left[ head^1 \oplus head^2 \oplus \ldots \oplus head^n \right] W_m \tag{3}$$

$$head^i = \text{Attention}^i(K, Q) \tag{4}$$

where "$\oplus$" represents the concatenation of vectors and $W_m \in R^{h_{dim}}$ is a learnable weight matrix. In the first attention layer, the semantic feature extraction module utilizes self-attention with $K = Q = H^c$ to learn the correlation within the sequence and obtain the introspective feature representation, $H^{cs}$. In addition, it employs interactive attention with $K = H^c$ and $Q = H^a$ to acquire the aspect-aware representation, $H^{as}$, with a primary focus on capturing the interactions between aspects and contexts. The second attention layer exploits a MHA with $K = H^{cs}$ and $Q = H^{as}$ to obtain a richer semantic representation, $H^{sem}$:

$$H^{cs} = \text{MHA}(H^c, H^c) \tag{5}$$

$$H^{as} = \text{MHA}(H^c, H^a) \tag{6}$$

$$H^{sem} = \text{MHA}(H^{cs}, H^{as}) \tag{7}$$

### 3.4. LDEGCN

3.4.1. Enhanced by Affective Knowledge

GCNs have the capability to directly operate on graph-structured data, making them widely used for encoding syntactic information [26]. To construct the local dependency-enhanced graph, we first construct the dependency graph for each input sentence over the dependency tree following the approach proposed in [11]. The dependency tree encapsulates lexical relationships and holds vital significance for understanding sentence structure and meaning. By extracting the connections between words from the dependency tree, we can more accurately capture the associations between emotional expressions and context, leading to better analysis and interpretation of the sentiment polarity of sentences. The adjacency matrix, $D \in R^{n \times n}$, of the sentence is obtained as follows:

$$D_{i,j} = \begin{cases} 1 & \text{if } w_i \text{ and } w_j \text{ have dependences} \\ 0 & \text{otherwise} \end{cases} \tag{8}$$

Inspired by Liang et al. [24], we incorporate the sentiment knowledge from SenticNet into the construction of the adjacency matrix to represent the emotional information between words. We utilize the latest version of SenticNet7 (http://sentic.net, accessed on 1 January 2023), which employs a novel commonsense-based neural–symbolic AI framework

for interpretable sentiment analysis and assigns semantic and emotional values to 400,000 concepts. Table 1 showcases examples of selected words and their corresponding sentiment intensities.

**Table 1.** Samples of affective words in SenticNet.

| Word | Intensity |
|---|---|
| Good | 0.659 |
| Excellent | 0.744 |
| Romantic | 0.851 |
| Reasonable | 0.170 |
| Bad | −0.659 |
| Horrible | −0.793 |

The calculation of specific sentiment scores is as follows:

$$I_{i,j} = |Intensity(w_i)| + |Intensity(w_i)| \tag{9}$$

The intensity score, $Intensity(w_i) \in [-1, 1]$, indicates the emotional intensity score of the word, $w_i$, in the SenticNet sentiment dictionary, with negative values indicating negative emotions, positive values indicating positive emotions and 0 indicating neutrality. The SenticNet sentiment dictionary comprises mappings of words to their corresponding emotional polarity and intensity. By simply matching each emotion word with SenticNet and extracting emotional intensity values from it, we can assign the intensity scores to the emotional words. For example, consider the sentence: "This product is excellent." When we analyze this sentence using SenticNet, the model looks up its emotional intensity score in SenticNet. Consequently, the word "excellent" in the sentence would be assigned an emotional intensity score of 0.744, conveying a positive sentiment. The graph convolutional network can extract sentiment dependencies from two dependent nodes, where the sentiment dependency correlation is determined by the sum of the absolute values of their sentiment intensity scores. By considering the strength of sentiment relationships between nodes, the model can understand the sentiment dependencies in the graph and analyze the sentiment associations between different aspects or entities.

### 3.4.2. Enhanced by Aspect and Dependency Types

Aspect terms are crucial in ABSA tasks; thus, it is necessary to further enhance the dependency relationships between aspects and contexts:

$$A_{i,j} = \begin{cases} 1 & \text{if } w_i \text{ or } w_j \text{ is a aspect word} \\ 0 & \text{otherwise} \end{cases} \tag{10}$$

To better identify context words or opinion words related to the aspect, we construct a specific set of dependency types, denoted as $S_{set}$ = {"nsubj", "dobj", "amod", "advmod", "neg", "acomp"}. When building the adjacency matrix, we consider the dependency types between words and enhance the specific dependency relationships. Let $R_{w_i,w_j}$ represent the dependency type between words, $w_i$ and $w_j$; if $R_{w_i,w_j} \in S_{set}$, we increase the weight of that dependency type in the graph:

$$T_{i,j} = \begin{cases} 1 & R_{w_i,w_j} \in S_{set} \\ 0 & \text{otherwise} \end{cases} \tag{11}$$

By utilizing the adjacency matrix, $D$; the aspect matrix, $A$; the sentiment matrix, $I$; and the type-enhanced matrix, $T$, we are able to derive the dependency-enhanced adjacency matrix, $U$:

$$U_{i,j} = D_{i,j} \times (I_{i,j} + A_{i,j} + T_{i,j} + 1) \tag{12}$$

### 3.4.3. Local Context Weight

Dependency syntactic structure can reveal the semantic modification relationships between different constituents of a sentence, enabling it to capture collocational information over long distances. The dependency graph constructed through dependency syntactic trees allows for the effective acquisition of long-distance dependencies among various words in a sentence. According to the literature [27], the sentiment polarity of aspects is primarily influenced by the surrounding contexts. As the distance between the context and the aspect increases, the correlation gradually weakens. This characteristic was not effectively addressed in previous works. Hence, the LCW method was introduced for improved handling of long-distance dependencies. This method assigns higher weights to words in close proximity while reducing the weights of words farther away. This guides the model to focus more on local correlations, enhancing its performance in addressing long-distance dependency issues.

The LCW method constructs a local context-weighted adjacency matrix and leverages the dependency parse tree to calculate the syntactic distance (*SD*) between contexts and aspects in order to assign position weights. The syntactic distances from contexts to different aspects are illustrated in Figures 3 and 4. Formally, the weight of dependency $(w_i, w_j)$ can be defined as follows:

$$W_{i,j} = \begin{cases} 1 - \frac{d_i + d_j}{2n} & d_i \text{ and } d_j \text{ exist} \\ 0 & \text{no connection} \end{cases} \tag{13}$$

where $d_i$, $d_j$ represent the syntactic distance in the dependency tree from the *i*-th word and the *j*-th word to the given aspect, respectively, and *n* represents the length of the current sentence.

$$G_{i,j} = U_{i,j} * W_{i,j} \tag{14}$$

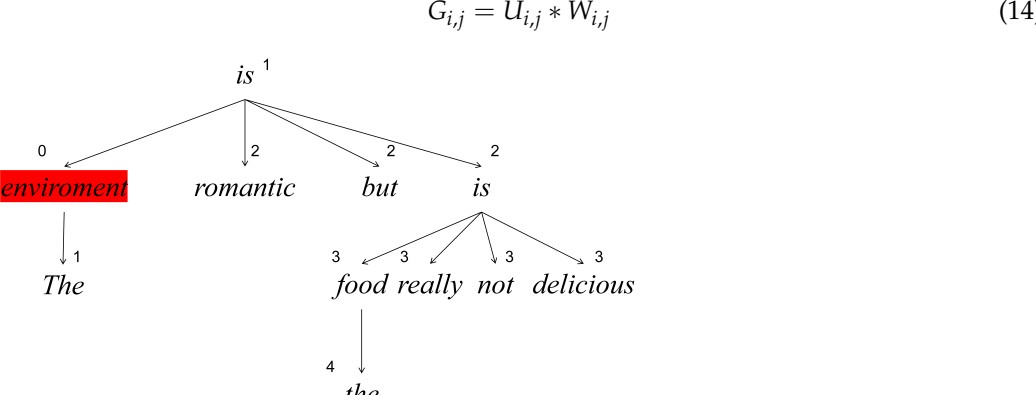

**Figure 3.** The positional relationship of the aspect "environment" in dependency tree (The number on each word represents the *SD* to the aspect "environment").

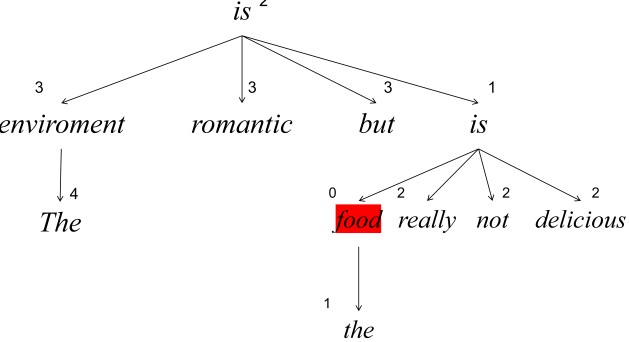

**Figure 4.** The positional relationship of the aspect "food" in dependency tree (The number on each word represents the *SD* to the aspect "food").

The dependency graph enhanced by LCW not only incorporates the syntactic information of the context but also effectively filters out noise caused by irrelevant information to the aspect. Based on the enhanced graph, the multi-layer GCN facilitates the efficient fusion of global information and local sentiment details. Algorithm 1 provides an overview of the procedure for generating the dependency-enhanced matrix for each sentence:

---

**Algorithm 1** The process of generating a dependency-enhanced matrix.

---

**Require:** a sentence, $W^c = \{w_1^c, w_2^c, \ldots, w_{a+1}^c, \ldots, w_{a+m}^c, \ldots, w_n^c\}$; aspect sequence, $W^a = \{w_1^a, w_2^a, \ldots, w_n^a\}$; the dependency tree of the sentence, $dependency(W^d)$; intensity scores from SenticNet; a specific dependency set, $S_{set}$; syntax distance, $SD = \{d_1, d_2, \ldots, d_n\}$

1:  **for** i = 1 → n **do**
2:      **for** j = 1 → n **do**
3:          **if** $dependency(w_i, w_j) \in dependency(W^d)$ or $i = j$ **then**
4:              $D_{i,j} \leftarrow 1$                    ▽Generated by dependency tree
5:              $I_{i,j} \leftarrow |Intensity(w_i)| + |Intensity(w_j)|$        ▽Enhanced by SenticNet
6:              **if** $w_i^c$ or $w_j^c \in W^a$ **then**
7:                  $A_{i,j} \leftarrow 1$
8:                  **if** $R_{i,j} \in S_{set}$ **then**
9:                      $T_{i,j} \leftarrow 1$            ▽Enhanced by dependency type
10:                  **else**
11:                      $T_{i,j} \leftarrow 0$
12:                  **endif**
13:              **else**
14:                  $A_{i,j} \leftarrow 0$
15:              **endif**
16:          **else**
17:              $D_{i,j} \leftarrow 0$
18:          **endif**
19:          $U_{i,j} \leftarrow D_{i,j} \times (I_{i,j} + A_{i,j} + T_{i,j} + 1)$
20:          $W_{i,j} \leftarrow 1 - \frac{d_i + d_j}{2n}$                ▽Calculating weight matrix(W)
21:          $G_{i,j} \leftarrow U_{i,j} \times W_{i,j}$
22:      **endfor**
23: **endfor**

---

### 3.4.4. Multilayer GCN

The contextual feature, $H^c$, from BERT, along with the dependency-enhanced graph serves as the input to the first GCN layer. The output of each GCN layer as well as the dependency-enhanced graph continue to be utilized as the new input to the next GCN layer. By sequentially propagating through $l$ layers of GCN, we obtain the final contextual feature representation, $H^l \in R^{n \times h_{dim}}$. In GCN, each node's representation is updated based on information from its neighboring nodes. At the $l$-th GCN layer, the state of each node is updated as follows:

$$h_i^l = ReLU(G_i h_i^{l-1} W^l + b^l) \tag{15}$$

$$H^l = \left\{ h_1^l, h_2^l, \ldots, h_n^l \right\} \tag{16}$$

### 3.5. Feature Fusion

To enrich the final feature representation, $h^p$, for classification, we perform mean pooling on the semantic feature, $H^{sem}$, the feature representation $H^l$ after $l$ layers of GCN

and contextual encoding, $H^c$. Subsequently, we concatenate these pooled representations together:

$$h_{avg}^{sem} = \frac{\sum_1^n h_i^{sem}}{n} \tag{17}$$

$$h_{avg}^{l} = \frac{\sum_1^n h_i^{l}}{n} \tag{18}$$

$$h_{avg}^{c} = \frac{\sum_1^n h_i^{c}}{n} \tag{19}$$

$$h^p = \left[ h_{avg}^{sem} \oplus h_{avg}^{l} \oplus h_{avg}^{c} \right] \tag{20}$$

### 3.6. Sentiment Classifier

Once we acquire the final feature representation, $h^p$, we input it into a fully connected layer and then perform softmax normalization to generate the probability distribution, $p \in R^{d_p}$, of different sentiment polarities:

$$p = \text{softmax}(W^p h^p + b^p) \tag{21}$$

where $d_p$ is the same as the dimension of the sentiment labels. $W_p \in R^{d_p \times d_h}$ and $b^p \in R^{d_p}$ are the learnable matrix and bias.

The parameters of our model are updated using the gradient descent algorithm. The objective of training the model is to minimize the cross-entropy loss with $L2$ regularization:

$$L(\theta) = \sum_{i=1}^{S} \sum_{j=1}^{C} \hat{p}_i^j \log\left( p_i^j \right) + \lambda \|\Theta\|^2 \tag{22}$$

where $S$ contains all sentence–aspect pairs, $C$ is the collection of sentiment polarities, $\hat{p}_i^j$ is the real distribution of sentiment, $\Theta$ represents all trainable parameters and $\lambda$ is the coefficient for the $L2$ regularization term.

## 4. Experiment

### 4.1. Datasets and Experiment Setting

To evaluate the generalizability of LDEGCN, we conducted experiments on the following five benchmark datasets: Twitter, consisting of Twitter posts [28] and restaurant and laptop reviews from SemEval 2014 Task 4 [29], SemEval 2015 task 12 [30] and SemEval 2016 task 5 [31]. All five datasets consist of three sentiment polarities: positive, negative and neutral. Each dataset includes sentences that are annotated with marked aspects and their corresponding polarities. Table 2 presents the statistical information for these datasets.

**Table 2.** Statistics of the datasets.

| Dataset | Type | Positive | Negative | Neural |
|---------|------|----------|----------|--------|
| Twitter | Train | 1561 | 3127 | 1560 |
| | Test | 174 | 346 | 173 |
| Lap14 | Train | 994 | 464 | 870 |
| | Test | 341 | 169 | 128 |
| Rest14 | Train | 2164 | 637 | 807 |
| | Test | 728 | 196 | 196 |
| Rest15 | Train | 912 | 36 | 256 |
| | Test | 326 | 34 | 182 |
| Rest16 | Train | 1240 | 69 | 439 |
| | Test | 469 | 30 | 117 |

In our experiments, we utilized the spacy (https://spacy.io, accessed on 1 December 2022) tool to construct the dependency syntactic tree of the given sentences. We employed the pre-trained BERT (https://github.com/huggingface, accessed on 1 August 2022) model to encode the given sentences and obtain word embeddings with an initial dimensionality of 768. For parameter settings, the BERT model was initialized with pre-trained parameters, while the remaining trainable parameters were initialized using the Xavier initialization method [32]. During the training, the model was trained with a batch size of 16 comments. We employed a learning rate of $2 \times 10^{-5}$ with a cross-entropy loss function. The Adam optimizer was utilized with a learning rate of 0.003. Dropout and early stopping techniques were applied to prevent overfitting.

### 4.2. Comparative Models

To evaluate the performance of LDEGCN, we compared it with the baseline models and many state-of-the-art models. The specific methods are as follows:

IAN [8] learns the relationship between contexts and aspects using an interactive attention network.

AOA [9] prioritizes the crucial parts of a sentence through the attention-over-attention module and extracts the interactive feature between context and aspect words.

BERT-SPC [32] inputs the sequence in the form of "[CLS] sentence [SEP] aspect [SEP]" into the pre-trained BERT model for prediction.

AEN-BERT [25] employs an attention encoder network to model the relationship between contexts and targets.

ASGCN [11] initially introduces the method of constructing an adjacency graph using the syntactic dependency tree of a sentence.

DGEDT-BERT [33] introduces an enhanced dual-transformer network to jointly examine flat representations and graph-based representations.

BERT4GCN [34] combines the intermediate outputs of BERT with the positional information between words to enhance the GCN encoding process.

T-GCN [17] utilizes a multiple-layer type-aware GCN for comprehensive learning of different edge relationships.

DualGCN [15] introduces a dual-graph model that simultaneously considers the syntactic structure and semantic relations. Furthermore, the SynGCN and SemGCN networks are integrated using a bidirectional BiAffine module.

SSEGCN [16] integrates the attention matrix and the syntactic mask matrix to facilitate the interaction between syntactic structure and semantic information.

### 4.3. Results and Analysis

To validate the effectiveness of LDEGCN, we compared it against previous benchmark methods. We adopt accuracy and macro-averaged F1 as evaluation metrics because accuracy is widely used in classification tasks and macro-averaged F1 scores are suitable for datasets with class imbalance.

Compared to traditional attention-based models, LDEGCN mitigates the noise introduced by the attention mechanism, which could misguide the model to learn information irrelevant to aspects. In contrast to GCN-based models, LDEGCN incorporates sentiment knowledge and weights the dependency items based on sentiment scores and dependency types, allowing for better utilization of syntactic dependency information.

Detailed results are presented in Table 3, which demonstrates the superior performance of LDEGCN. Across the four SemEval task datasets, LDEGCN outperforms most of the previous models and achieves near-optimal or even optimal results on some datasets. This highlights the model's ability to effectively leverage syntactic dependency information to extract ample syntactic and semantic features for sentiment classification. Regrettably, LDEGCN does not perform as well on the Twitter dataset as it does on other datasets. This could be due to the incomplete and colloquial nature of comments on Twitter, making it challenging for the dependency tree to accurately parse the sentence components. Never-

theless, the strong performance on other datasets still affirms the effectiveness of LDEGCN for ABSA.

**Table 3.** Experimental results on five datasets. (Acc represents accuracy, F1 represents Macro-F1 score. The best results are displayed in bold, the second-best results are underlined.)

| Model | Twitter | | Lap14 | | Rest14 | | Rest15 | | Rest16 | |
|---|---|---|---|---|---|---|---|---|---|---|
| | Acc | F1 | Acc | F1 | Acc | F1 | Acc | F1 | Acc | F1 |
| IAN | 72.50 | 70.81 | 72.05 | 67.38 | 79.26 | 70.09 | 78.54 | 52.65 | 84.74 | 55.21 |
| AOA | 72.30 | 70.20 | 72.62 | 67.52 | 79.97 | 70.42 | 78.17 | 57.02 | 87.50 | 66.21 |
| BERT-SPC | 75.92 | 75.18 | 77.59 | 75.03 | 84.11 | 76.68 | 83.48 | 66.18 | 90.10 | 74.16 |
| AEN-BERT | 74.54 | 73.26 | 79.93 | 76.31 | 83.12 | 73.76 | 82.29 | 63.41 | 88.96 | 70.31 |
| ASGCN | 72.15 | 70.40 | 75.55 | 71.05 | 80.77 | 72.02 | 79.89 | 61.89 | 88.99 | 67.48 |
| DGEDT | **77.90** | 75.40 | 79.80 | 75.60 | 86.30 | <u>79.89</u> | 84.00 | 71.00 | <u>91.90</u> | <u>79.00</u> |
| BERT4GCN | 74.73 | 73.76 | 77.49 | 73.01 | 84.75 | 77.11 | - | - | - | - |
| T-GCN | <u>76.45</u> | <u>75.25</u> | 80.88 | 77.03 | 86.16 | 79.95 | <u>85.26</u> | <u>71.69</u> | **92.32** | 77.29 |
| DualGCN | 77.40 | **76.02** | **81.80** | <u>78.10</u> | <u>87.13</u> | **81.16** | - | - | - | - |
| SSEGCN | 77.40 | **76.02** | 81.01 | 77.96 | **87.31** | 81.09 | - | - | - | - |
| Our model | 76.43 | 75.22 | <u>81.25</u> | **78.17** | 86.34 | **81.16** | 85.42 | 72.05 | 91.56 | **79.45** |

*4.4. Ablation Study*

To investigate the effectiveness of each module in LDEGCN, extensive ablation experiments were conducted. The results are presented in Table 4, where "w/o" denotes "without". The ablation experiments clearly demonstrate that excluding any component from the complete model leads to a degradation in performance. This underscores the indispensability of all components, as each one contributes uniquely to the model's performance.

**Table 4.** Ablation experiment.

| Model | Twitter | | Lap14 | | Rest14 | | Rest15 | | Rest16 | |
|---|---|---|---|---|---|---|---|---|---|---|
| | Acc | F1 | Acc | F1 | Acc | F1 | Acc | F1 | Acc | F1 |
| W/o SEM | 75.73 | 74.10 | 80.13 | 76.81 | 85.62 | 80.07 | 84.87 | 71.42 | 90.8 | 77.65 |
| W/o SenticNet | 75.02 | 73.07 | 80.09 | 76.64 | 85.09 | 77.96 | 83.23 | 68.54 | 89.98 | 76.71 |
| W/o type | 75.43 | 74.22 | 80.88 | 77.52 | 85.26 | 78.74 | 84.56 | 70.88 | 90.59 | 77.45 |
| W/o LCW | 76.16 | 74.94 | 80.41 | 77.12 | 85.68 | 79.20 | 84.98 | 71.24 | 91.07 | 78.37 |
| W/o LDEGCN | 74.49 | 72.44 | 79.62 | 75.65 | 84.82 | 77.14 | 82.29 | 67.59 | 88.97 | 73.26 |

It can be observed that (1) removing the semantic feature extraction module (w/o SEM) resulted in a noticeable drop in performance, indicating that the semantic feature extraction module plays a crucial role in reducing noise and learning the presentation of aspect-related semantic features; (2) the removal of SentiNet (w/o SenticNet) and dependency type (w/o type) both resulted in a decrease in model performance, indicating that they both assist the model in focusing on more informative dependencies. External sentiment knowledge provides additional information about emotions to better understand and express sentiment-related aspects. Meanwhile, specific sets of dependencies aid the model in emphasizing crucial dependency relationships when modeling connections between words; (3) the removal of the LCW method leads to a slight decline in model performance, suggesting that utilizing the local context weight method on the graph can reduce attention to irrelevant features and alleviate issues related to long-distance dependencies; and (4) the significant degradation in performance was observed when the entire LDEGCN module was removed underscores the effectiveness of LDEGCN in providing sentiment information to the dependencies between context and aspect. This enables the model to obtain more accurate sentiment features and enhance the prediction performance for aspect-specific sentiment.

Overall, our ablation study provides compelling evidence for the effectiveness of each module in the LDEGCN model, demonstrating their unique contributions to achieving superior performance in sentiment analysis tasks.

## 5. Discussion

### 5.1. The Influence of the Number of Layers (L)

Selecting an appropriate number of GCN layers (L) is beneficial for model performance, so we explored the impact of different L values, as shown in Figure 5. The results indicate that when $L = 2$, the Acc and F1 scores are the highest across all datasets.

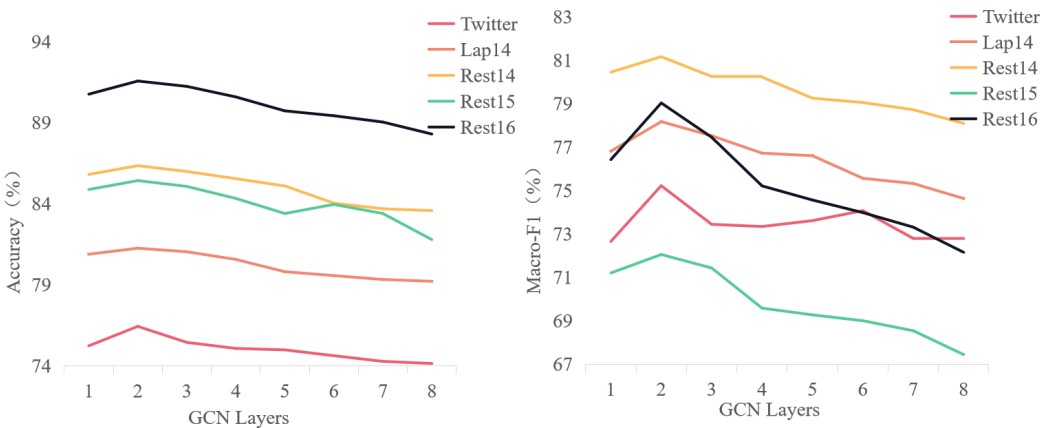

**Figure 5.** Impact of GCN layer, L, on the model effect.

The aspect "food" has different syntactic distances from the context. With a single GCN layer, each word node aggregates feature information from adjacent word nodes. Therefore, the aspect "food" captures feature information only from the context with an *SD* of 1. When *L* is 1, it is evident that "the" and "is" do not provide sufficient information. With two GCN layers, "food" captures important sentiment features from the context words "really", "not" and "delicious" with an *SD* of 2. However, if *L* is set to 3 or more, it introduces irrelevant feature information, such as the positive sentiment feature from the word "romantic".

In summary, when $L = 1$, our model fails to capture sufficient aspect-contextual feature information, which may lead to incorrect judgments. When $L = 2$, the aspect "food" itself has high semantic importance, and the semantic importance of "really", "not", and "delicious" increases, which benefits our model. However, when $L = 3$ or more, the distribution of semantic importance in the context becomes dispersed, making it difficult for the aspect "food" to focus on important information. A visualization of the attention distribution from Layer 1 to Layer 3 is shown in Figure 6.

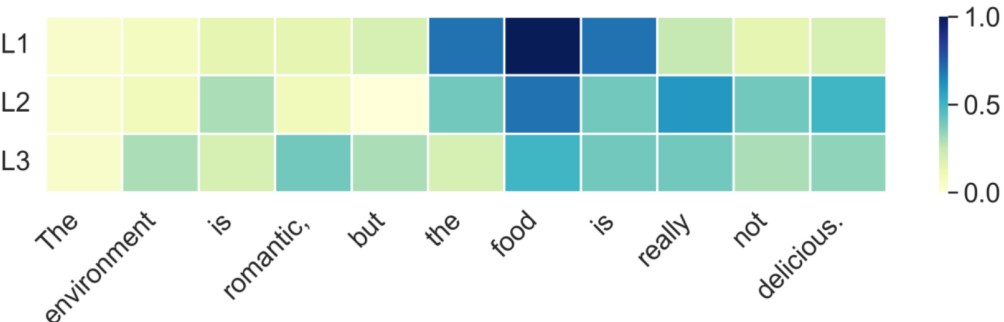

**Figure 6.** Visualization of semantic weights for different L.

### 5.2. The Impact of Sentiment Words with Different Scores

To investigate the impact of sentiment words with different intensity levels on model performance, we sorted all sentiment words in descending order based on the absolute values of their sentiment scores. We constructed the dependency-enhanced graphs using an equal number of sentiment words from different score ranges and conducted experiments on the Rest14 dataset. The result is shown in Figure 7, revealing that using sentiment words with higher scores to construct the LDE graph yielded better performance. This indicates that sentiment words with stronger polarity contribute to higher word correlations within sentences, resulting in more accurate derived word correlation graphs. In other words, the stronger the sentiment tendency of the sentiment words, the stronger the word correlations in the sentence, leading to more precise word correlation graphs.

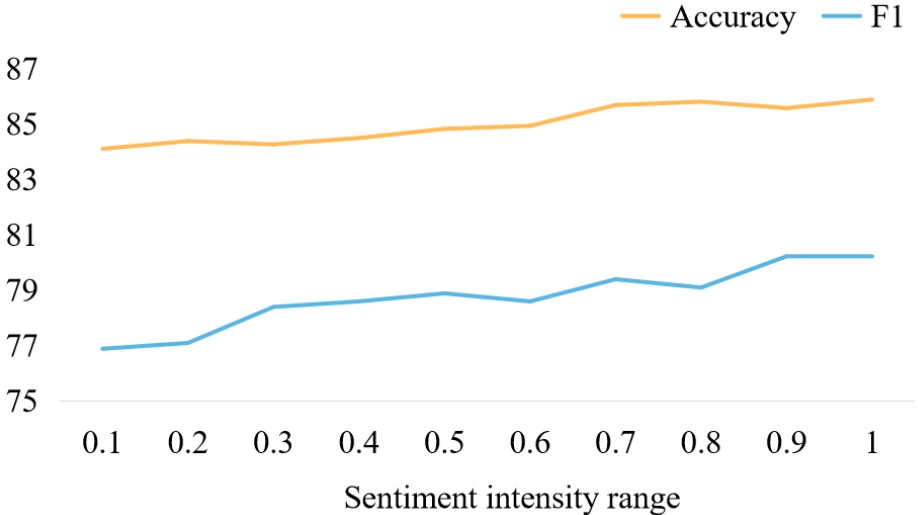

**Figure 7.** The impact of sentiment words with different scores.

### 5.3. Visualization of LDEG

To validate the effectiveness of LDEG, we present heatmaps to visually illustrate the construction process from a standard dependency graph to a local dependency-enhanced graph. The intensity of colors in the heatmaps represents the correlation between different words, with darker colors indicating stronger correlations. In Figure 8a, an adjacency matrix constructed from the syntactic dependency tree is utilized to visualize the dependency relationships between nodes. However, the equal treatment of these dependency edges hinders the identification of their relative importance. To overcome this limitation, we incorporate external sentiment knowledge, as depicted in Figure 8b, enabling the model to focus on words with prominent sentiment tendencies. Recognizing the significance of aspects in sentiment analysis tasks, we enhance the dependency edges directly connected to the aspects. Additionally, considering the varying importance of different dependency types, we further enhance attention toward dependencies within specific sets, as illustrated in Figure 8c. Nevertheless, we observe that the dependency relationship between "is" and "romantic" receives excessive attention, which is undesirable since "romantic" does not modify the aspect term "food". To address this issue, we employ the LCW method on the graph. As demonstrated in Figure 8d, the attention between "is" and "romantic", as well as "is" and "environment", is reduced. This allows the model to effectively aggregate feature information from important contextual words, ensuring focus on the relevant and informative aspects of the input text.

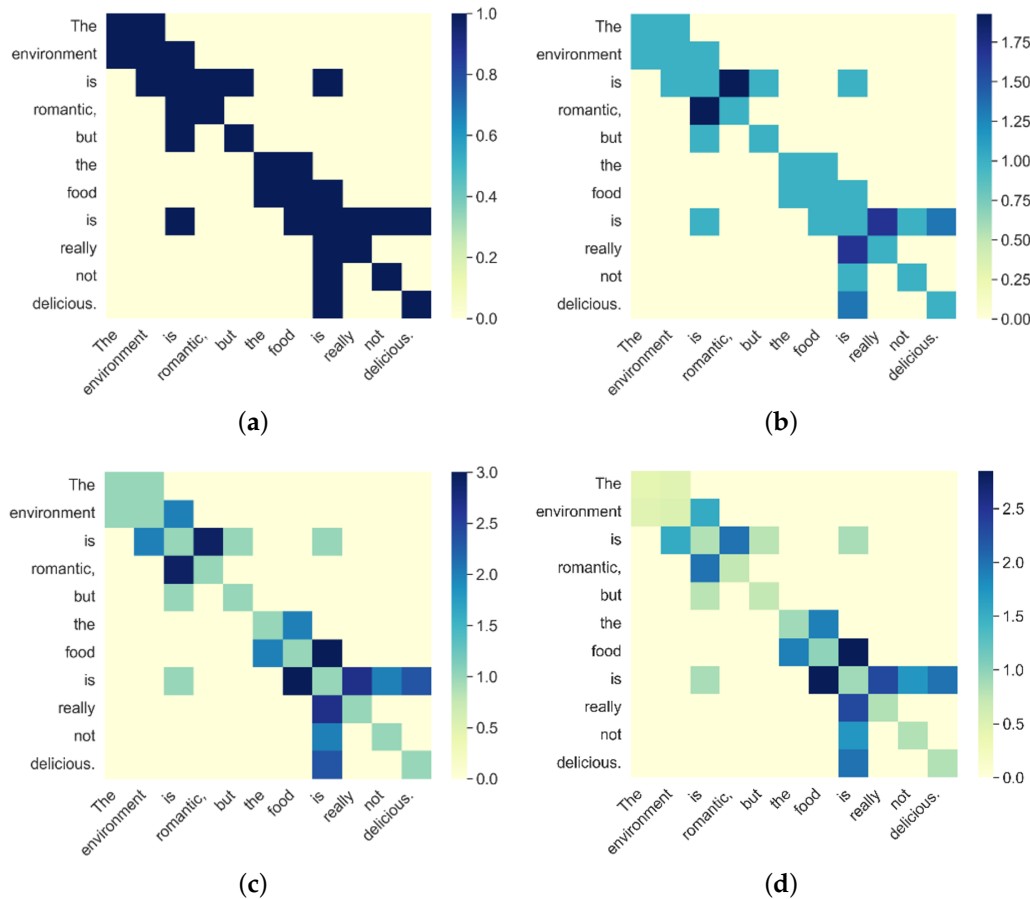

**Figure 8.** Visualization of LDE. (**a**) Dependency graph; (**b**) enhanced by SenticNet; (**c**) enhanced by aspect and dependency type; (**d**) enhanced by LCW.

## 6. Conclusions and Future Work

In this study, we propose a novel model LDEGCN that combines external sentiment knowledge and dependency types to address the issues of noise introduced by traditional attention mechanisms and the inefficient utilization of syntactic dependency trees. LDEGCN achieves second-best accuracy and the best F1 score on the "Lap14" dataset. There are significant improvements in the "Rest14", "Rest15" and "Rest16" datasets as well, with notable advancements in the "Rest15" dataset. The accuracy and F1 score both reach their best values, surpassing the second-best results by 0.16% and 0.36%, respectively. This success can be attributed to the collaborative impact of various modules in our model. The "SEM" module enhances semantic feature extraction through multi-head interactive attention and multi-head self-attention mechanisms. Additionally, SenticNet and the specific dependency set, $S_{set}$, increase the focus on dependencies carrying more sentiment information in the dependency graph. The LCW method, building upon the dependency-enhanced graph, augments the attention toward local dependency elements. Lastly, the feature fusion module combines extracted syntactic and semantic features with overall semantic features to acquire more comprehensive hierarchical features.

In future research, our primary goal is to enhance the model's performance when handling insufficient syntactic dependency parsing, as text data in real-world applications are often not well-formed. Additionally, we plan to evaluate the generalization performance of our model by applying it to different domains and assessing its performance on multilingual datasets. We will persist in developing methods that can explore deeper interrelationships between aspects and uncover more profound and nuanced connections between syntactic and semantic information.

**Author Contributions:** Conceptualization, F.W. and X.L.; methodology, F.W.; validation, F.W.; investigation, F.W.; writing—original draft preparation, F.W.; writing—review and editing, F.W. and X.L.; visualization, F.W. All authors have read and agreed to the published version of the manuscript.

**Funding:** This research received no specific grant from any funding agency in the public, commercial or not-for-profit sectors.

**Institutional Review Board Statement:** Not applicable.

**Informed Consent Statement:** Not applicable.

**Data Availability Statement:** Not applicable.

**Conflicts of Interest:** The authors declare no conflict of interest.

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
