# Peer review of "Local Dependency-Enhanced Graph Convolutional Network for Aspect-Based Sentiment Analysis"

_applsci, doi:10.3390/app13179669_

Round 1

Reviewer 1 Report

Some domain-specific terms, like "semantic feature extraction module" and "dependency-enhanced module," are used without prior explanation. Consider providing a brief description or context for these terms to assist readers who may not be familiar with the specific jargon.

 In the "local context weight (LCW) method" description, you mention "effectively solving the problem of long-distance dependencies." Consider providing a concise explanation of what the long-distance dependency problem is and how the LCW method mitigates it.

Consider adding a sentence or two that explains the significance of learning from dependency trees and how it relates to capturing contextual information.

Consider adding a brief introductory paragraph at the beginning of the "Proposed approach/Methodology" section to provide a high-level overview of the LDEGCN model. 

Explain how sentiment intensity scores are assigned based on the sentiment lexicon, and provide an example or two to illustrate the process.

some minor revision for simplicity of text is required

Reviewer 2 Report

Excelent works.

Just a few questions.

186 We utilize the latest version of SenticNet: SenticNet 7 [Refx1]  which employs a novel commonsense-based …

Ref x1Cambria, E., Liu, Q., Decherchi, S., Xing, F., & Kwok, K. (2022). SenticNet 7: A commonsense-based neurosymbolic AI framework for explainable sentiment analysis. In Proceedings of the Thirteenth Language Resources and Evaluation Conference (pp. 3829-3839).

I can't find the original reference either.

Tian et al. [17] argue 47 that dependencies of the "nsubj" type are more important than dependencies of the "det" 48 and "compound" types, and they have provided evidence to support this claim.

A brief explanation of the dependencies ("nsubj" , "det", etc) would help to clarify the paper.

27 Consequently, ABSA has emerged as a crucial research area in the field of Natural

Language Processing (NLP). [3,4].

31 each aspect.[5].

39 corresponding opinion words [11–16]. tThese …

verify that in the rest of the text the brackets of the references are preceded by a space.

8 Dependency-Enhanced Graph Convolutional Network (LDEGCN). (add space)

284 4.2. cComparative models

Table 1. Samples of affective words in SenticNet

Xavier initialization method. [32]. D

reasonable 0.170 (complet to 3 decimals)

Table 3.

Ttwitter Lap14 rRest14 Rest15 rRest16

Round al number to 2 decimals

e.g.

AEN-BERT 74.54

ASGCN 72.20

DGEDT 77.90

281 The Adam optimizer [¿?] was utilized to iteratively optimize …

Reviewer 3 Report

Article: Local dependency-enhanced graph convolutional network for aspect-based sentiment analysis

1.       Has this new method validated against Golden standards?  Comparison of results may be shown in conclusion part.

2.       Table: 2 – How Train data set and Test data set chosen may be explained (why 90% train and 10% test approximately)

3.       Table :3 – 80% + accuracy in proposed model. Discuss reasons for these accuracies in DISCUSSION / CONCLUSION part.

4.       Author could have tried with other computational methods such as SVM and compared with CNN. Any specific reasons for not selecting?

5.       Few of the References not discussing about computational methods. Seems not relevant.
